# ACTIVE AUDIO CANCELLATION WITH MULTI-BAND MAMBA NETWORK

## ABSTRACT

A novel deep learning approach for Active Audio Cancellation (AAC) is presented, which extends the capabilities of traditional Active Noise Cancellation (ANC) by addressing a wider range of audio signals, including those with complex spectral content. We propose, for the first time, a deep learning approach to AAC using a novel multi-band Mamba architecture. This architecture partitions input audio into multiple frequency bands, allowing for precise anti-signal generation and enhanced phase alignment across frequencies, thereby improving overall cancellation performance. Additionally, we introduce an optimization-driven loss function that provides near-optimal supervisory signals for anti-signal generation. Our experimental results demonstrate substantial improvements over existing methods, achieving up to 7.2dB gain in ANC scenarios and up to 6.2dB improvement in AAC for voice audio signals, outperforming existing methods.

## 1 INTRODUCTION

Active Noise Cancellation (ANC) is a critical audio processing technique aimed at eliminating unwanted noise by generating an anti-noise signal (Lueg, 1936; Hansen et al., 1997; Fuller et al., 1996; Kuo & Morgan, 1999; Nelson & Elliott, 1991). ANC has practical applications in improving hearing devices for individuals with hearing impairments and reducing chronic noise exposure, thereby mitigating hearing loss risks. It also enhances focus, productivity, and listening experiences while reducing stress. Traditional ANC algorithms, like LMS and its deep learning variants (Zhang & Wang, 2021; Park et al., 2023; Mostafavi & Cha, 2023; Cha et al., 2023; Singh et al., 2024; Pike & Cheer, 2023), have been widely adopted. However, these methods face limitations when dealing with more complex and high-frequency audio signals, as they are primarily designed to target noise.

This paper addresses the more general problem of Active Audio Cancellation (AAC), which extends beyond traditional noise cancellation to encompass the cancellation of any audio signal, irrespective of its spectrum. While ANC systems may implicitly aim to cancel any incoming sound, including speech, their primary focus has historically been on noise. Our work represents, to the best of our knowledge, the first attempt to actively cancel general audio signals with deep learning. This distinction opens new research avenues, as AAC does not rely on prior assumptions about the input signal, making it inherently more complex and versatile.

Our results indicate the strong potential of generative neural networks in addressing both AAC and traditional ANC tasks. To this end, we introduce a novel multi-band Mamba architecture. This architecture is effective in real-world environments with diverse audio frequencies. By partitioning the input into frequency bands, the model enables precise control over anti-signal generation, improving phase alignment and cancellation performance. Additionally, an optimization-driven loss function provides near-optimal supervisory signals for the generation of anti-signals, resulting in superior performance in complex and dynamic acoustic scenarios. In dynamic and complex acoustic settings, this multi-band approach leads to substantial improvements over the existing methods, achieving up to 7.2 dB gain in ANC scenarios and a 6.2 dB improvement in AAC for voice audio signals. These results surpass the performance of existing deep learning-based baselines, which are considered state-of-the-art in the field.

## 2 RELATED WORK

**Active Noise Cancellation:** The concept of ANC was first introduced by Lueg Lueg (1936), who focused on the cancellation of sound oscillations. Given that ANC algorithms (Hansen et al., 1997; Fuller et al., 1996; Kuo & Morgan, 1999; Nelson & Elliott, 1991) must adapt to variations in amplitude, phase, and the movement of the noise source , most ANC algorithms are based on the Least Mean Squares (LMS) algorithm (Burgess, 1981) which has demonstrated effectiveness in echo cancellation. The FxLMS (Filtered-x LMS) algorithm extends the LMS approach to ANC by employing an adaptive filter that accounts for distortions in the primary path $P(z)$ and secondary path $S(z)$. Boucher et al. (1991) analyze the error introduce in the FxLMS algorithm due to inaccuracies in estimating the secondary path inverse $\hat{S}(z)$. The secondary path in adaptive filtering systems often introduces nonlinear distortions that degrade the performance the FxLMS algorithm. Several approaches have been proposed to mitigate these issues. The Filtered-S LMS (FSLMS) algorithm (Das & Panda, 2004) utilizes a single-layer Functional Link Artificial Neural Network (FLANN) (Patra et al., 1999) to address nonlinear distortions. Another approach, the Volterra Filtered-x LMS (VFXLMS) algorithm (Tan & Jiang, 2001), employs a multichannel structure for feedforward active noise control to better handle nonlinearity. The Bilinear FxLMS algorithm (Kuo & Wu, 2005) incorporates bilinear filters that offer an improved modeling of nonlinearity compared to the VFXLMS method. Additionally, the Leaky FxLMS (Tobias & Seara, 2005) algorithm introduces a "leakage" term in the coefficient updates, which helps mitigate overfitting to noise or rapid signal changes. The Tangential Hyperbolic Function-based FxLMS (THF-FxLMS) (Ghasemi et al., 2016) employs a tangential hyperbolic function to model the saturation effects of the loudspeaker, further enhancing performance in the presence of nonlinearities. Gannot & Yeredor (2003) propose blind source separation methods based on second-order statistics for noise cancellation. Moreover, Oppenheim et al. (1994) proposed single channel ANC based on Kalman filter formulation (Revach et al., 2021). Additionally, Rafaely (2009) investigate spherical loudspeaker arrays for local sound control, analyzing the interaction of primary and secondary sound fields to form shell-shaped quiet zones.

ANC using deep learning was first proposed by Zhang & Wang (2021), utilizing a convolutional-LSTM network to estimate both the amplitude and phase of the canceling signal $y(t)$. Similar approaches using recurrent CNNs were presented by Park et al. (2023), Mostafavi & Cha (2023) and by Cha et al. (2023). Furthermore, autoencoder-based networks have been utilized to address the ANC problem Singh et al. (2024), as well as fully connected neural networks Pike & Cheer (2023). Moreover, Shi et al. (2020; 2022b; 2023a), Luo et al. (2022), and Park & Park (2023) have developed methods that select fixed-filter ANC (SFANC) from pre-trained control filters to achieve fast response times. Furthermore, Luo et al. (2023b;a; 2024c) focused on generating filters for selective fixed-filter ANC. In parallel, Zhang & Wang (2023), Shi et al. (2024; 2023b), Antoñanzas et al. (2023), Xiao et al. (2023), Zhang et al. (2023b), and Zhu et al. (2021) contributed to the development of multichannel ANC systems. To address the challenges of real-time ANC, Luo et al. and Shi et al. proposed a convolutional neural network-based approach (Luo et al., 2024b; Shi et al., 2022a), which was later enhanced by integrating convolutional neural networks with Kalman filtering (Luo et al., 2023c). Additionally, Zhang et al. (2023a) introduced an attention mechanism for real-time ANC, leveraging the Attentive Recurrent Network (ARN) network (Pandey & Wang, 2022). Other notable contributions to real-time ANC include attentive recurrent networks (Zhang et al., 2022). Other innovative approaches include a genetic algorithm-based method for ANC proposed by Zhou et al. (2023) and a bee colony algorithm for ANC introduced by Ren & Zhang (2022).

**Active Speech Cancellation:** Active speech cancellation (ASC) has been explored in various studies, each employing different approaches to predict and cancel unwanted speech signals. Kondo & Nakagawa (2007) introduced an ASC method using a Linear Predictive Coding (LPC) model to predict the speech signal for generating the cancelling signal $y(t)$. Donley et al. (2017) took a different approach by controlling the sound field to cancel speech using a linear dipole array of loudspeakers and a single microphone, effectively reducing the speech signal in the target area. Iotov et al. (2022) employed a long-term linear prediction filter to anticipate incoming speech, enabling the cancellation of the speech signal. Additionally, Iotov et al. (2023) proposed the HOSpLP-ANC method, which utilizes an adaptive high-order sparse linear predictor alongside the Least Mean Squares (LMS) algorithm to achieve effective speech cancellation.

**Mamba architecture:** Recently, the Mamba architecture has been introduced (Gu & Dao, 2023; Dao & Gu, 2024), leveraging State Space Models (SSMs) to achieve notable improvements in various audio-related tasks. One of the key advantages of the Mamba architecture is its ability to perform fast inference, especially when handling sequences up to a million in length, which represents a significant improvement over traditional generative architectures. This has enabled advancements in several applications, including automatic speech recognition (Zhang et al., 2024b;a), speech separation (Jiang et al., 2024a; Li & Chen, 2024), speech enhancement (Chao et al., 2024; Luo et al., 2024a; Quan & Li, 2024), speech super-resolution (Lee & Kim, 2024), sound generation (Jiang et al., 2024b), audio representation (Shams et al., 2024; Yadav & Tan, 2024; Erol et al., 2024), sound localization (Xiao & Das, 2024; Mu et al., 2024), audio tagging (Lin & Hu, 2024), and deepfake audio detection (Chen et al., 2024).

## 3 APPROACH

### 3.1 BACKGORUND

The signal processing framework of a typical feedforward ANC system is detailed, emphasizing the roles of the primary and secondary acoustic paths. In such systems, reference and error microphone signals are utilized to generate a canceling signal that minimizes unwanted noise. The primary path $P(z)$ represents the acoustic transfer function from the noise source to the error microphone, while the secondary path $S(z)$ represents the acoustic transfer function from the loudspeaker to the error microphone. The signal captured by the reference microphone is denoted as $x(n)$, while the signal captured by the error microphone is denoted as $e(n)$. These signals are fed into the ANC controller, which processes them to produce a canceling signal $y(n)$. The canceling signal is then played through a loudspeaker, referred to as $f_{LS}$, producing $f_{LS}\{y(n)\}$, which aims to suppress the unwanted noise near the error microphone. The loudspeaker output $f_{LS}\{y(n)\}$, after passing through the secondary path $S(z)$, generates the anti-signal denoted by $a(n)$. The equation representing the relationship is:

$$a(n) = S(z) * f_{LS}\{y(n)\} \tag{1}$$

Similarly, the reference signal $x(n)$, transmitted through the primary path $P(z)$, produces the primary signal denoted by $d(n)$, which is defined as:

$$d(n) = P(z) * x(n) \tag{2}$$

The error signal $e(n)$, which represents the difference between the primary signal $d(n)$ and the anti-signal $a(n)$, is expressed as:

$$e(n) = d(n) - a(n) \tag{3}$$

The goal of the ANC controller is to minimize the error signal $e(n)$, ideally to zero, indicating successful noise cancellation. In the feedback ANC approach, only the error signal $e(n)$ is utilized to generate the canceling signal, focusing on minimizing the residual noise detected by the error microphone.

One of the widely used metrics for measuring noise attenuation in ANC is the Normalized Mean Square Error (NMSE) between two signals, defined by:

$$\text{NMSE}\,[\mathbf{u}, \mathbf{v}] = 10 \cdot \log_{10}\left(\frac{\sum_{n=1}^{M}(u(n) - v(n))^2}{\sum_{n=1}^{M} u(n)^2}\right) \tag{4}$$

where $\mathbf{u}$ and $\mathbf{v}$ are the vector representations of the signals $u(n)$ and $v(n)$ such that $\mathbf{u} = [u(1), ..., u(M)]$ and $\mathbf{v} = [v(1), ..., v(M)]$. Here, $M$ represents the total number of samples. Typically, $u(n)$ refers to the target signal, while $v(n)$ denotes the estimated signal. A lower NMSE value indicates a better estimation, reflecting a closer alignment between the estimated signal and the target signal. In the context of ANC, typically $u(n)$ is the primary signal $d(n)$, while $v(n)$ will be the anti-signal $a(n)$. A schematic representation of the ANC system is illustrated in Fig. 2.

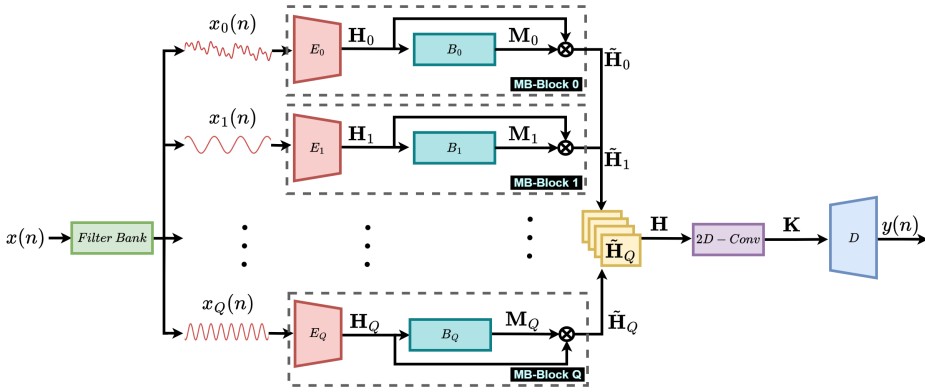

Figure 1: DeepAAC Architecture: the reference signal undergoes decomposition through a filter bank, dividing it into multiple frequency bands. Each band is processed by an encoder followed by a Mamba-based masking network. The resulting outputs from all frequency bands are concatenated and passed through a decoder to reconstruct the signal.

## 3.2 METHOD

The proposed method utilizes a novel architecture that integrates the Mamba framework (Gu & Dao, 2023) for the generation of the anti-signal. The architecture includes a filter bank that decomposes the input signal into multiple frequency bands, with each band processed by an encoder and a masking Mamba network. The outputs of the multi-band masking are then concatenated and passed through a decoder. Furthermore, we introduce a new loss function that leverages the near-optimal anti-signal as the ground truth, significantly improving the precision of the anti-signal generation process. A diagram of the proposed architecture is shown in Fig.1.

## 3.3 DEEPAAC ARCHITECTURE

Let $x(n)$ be the reference signal such that $1 \leq n \leq M$. The reference signal $x(n)$ is decomposed into $Q \in \mathbb{N}$ different frequency bands $x_1(n), \ldots, x_Q(n)$. These frequency bands are evenly divided such that for the maximum frequency $F$, the $i$-th frequency band $x_i(n)$ covers the frequency range $\left[(i-1)\frac{F}{Q}, i\frac{F}{Q}\right]$ where $1 \leq i \leq Q$. In addition to the decomposed bands, the original full-band signal $x(n)$ is included as $x_0(n)$. Each band $x_i(n)$ (where $0 \leq i \leq Q$, the zero index is for the entire unfiltered band) is then processed through its own Mamba-Band block (MB-block). Each MB-block comprises an encoder and a masking network that utilize Mamba-based layers. Within each MB-block, the encoder consists of a one-dimensional convolution layer $E_i$ with a kernel size $k$ and a stride of $k/2$. The encoder transforms the $i$-th reference signal $x_i(n)$ into a two-dimensional latent representation:

$$\mathbf{H}_i = E_i[\mathbf{x}_i] \tag{5}$$

where $\mathbf{H}_i \in \mathbb{R}^{B \times C}$, with $B = \frac{M-k}{\frac{k}{2}}+1$, $C$ representing the number of channels after the convolution operator and $\mathbf{x}_i$ is the vector representation of $x_i(n)$ . The latent representation $\mathbf{H}_i$ is then passed through the Mamba-based layers $B_i$ to produce the $i$-th masking signal $\mathbf{M}_i$ :

$$\mathbf{M}_i = B_i[\mathbf{H}_i] \tag{6}$$

The MB-blocks estimates $Q+1$ masks of the same latent dimension $\mathbf{M}_i \in \mathbb{R}^{B \times C}$. These masks are element-wise multiplied with the encoder outputs $\mathbf{H}_i$ to produce masked hidden representations $\tilde{\mathbf{H}}_i$:

$$\tilde{\mathbf{H}}_i = \mathbf{H}_i \cdot \mathbf{M}_i \tag{7}$$

Then, the masked hidden representations $\tilde{\mathbf{H}}_i$ is concatenated over all frequency bands $i$, such that:

$$\mathbf{H} = concat\left[\tilde{\mathbf{H}}_0, ..., \tilde{\mathbf{H}}_Q\right] \tag{8}$$

Where $\mathbf{H} \in \mathbb{R}^{(Q+1) \times B \times C}$. The hidden tensor $\mathbf{H}$ is then processed with a 2D convolution layer with a kernel size of $1 \times 1$ and one output channel that produces $\mathbf{K} \in \mathbb{R}^{B \times C}$. To obtain the vector representation of the canceling signal $\mathbf{y}$, we apply a decoder $D$. Specifically, the decoder is a one-dimensional transpose convolutional layer with a kernel size $k$ and a stride of $k/2$. This decoder ensures that the canceling signal $\mathbf{y}$ has the same dimensions as the reference signal $x(n)$:

$$\mathbf{y} = D[\mathbf{K}], \tag{9}$$

where $\mathbf{y} = [y(1), \ldots, y(M)]$ is the vector representation of the canceling signal $y(n)$, and $M$ is the length of the signal.

### 3.4 Optimization Objective

The training protocol for the proposed method consists of two distinct phases: (i) ANC Loss minimization, and (ii) Near Optimal Anti-Signal Optimization. Each phase employs the NMSE loss function (Eq. 4) but with different optimization objectives.

**ANC Loss:** In the first phase, the optimization aims to minimize the residual error signal. Given a reference signal $x(n)$ and the model output $y(n)$, the error loss function is defined as follows:

$$\mathcal{L}_{\text{ANC}} = \text{NMSE}\left[\mathbf{P} * \mathbf{x}, \mathbf{S} * f_{LS}\{\mathbf{y}\}\right] \tag{10}$$

where $\mathbf{P}$ and $\mathbf{S}$ represent the vectorized forms of the primary-path impulse response $P(z)$ and the secondary-path impulse response $S(z)$, respectively; $\mathbf{x}$ and $\mathbf{y}$ are the vectorized forms of the reference signal $x(n)$ and the canceling signal $y(n)$. The operator $*$ denotes convolution. Both $\mathbf{P}$ and $\mathbf{S}$ are obtained from the simulator employed in our study.

**Near Optimal Anti-Signal Optimization (NOAS):** One of the primary challenges in formulating ANC as a supervised learning problem lies in defining an appropriate training objective that accounts for the characteristics of the secondary path $S(z)$ and the primary path $P(z)$. In an ANC algorithms, the output $y(n)$ is processed by a nonlinearity function $f_{LS}$ and then propagated through the secondary path $S(z)$. The training objective aims to minimize the error signal $e(n)$, which represents the residual noise after cancellation.

However, this process becomes problematic when the secondary path $S(z)$ attenuates certain frequencies that are present in the primary signal $d(n)$. Under the vanilla loss function (e.g., Eq. 11), the model can be unfairly penalized for high error signals in these attenuated frequency bands, even when it has generated an optimal anti-signal. This occurs because the secondary path inherently suppresses these frequencies, leading to residual energy in the error signal $e(n)$. As a result, the training process encounters discrepancies that hinder the model's ability to learn effectively.

To address this challenge, we propose the NOAS optimization loss function (Eq. 12). The NOAS loss symmetrically incorporates the secondary path $S(z)$ on both sides of the NMSE calculation. By doing so, it ensures that any frequencies nullified by $S(z)$ are also excluded from the target, thereby mitigating the contribution of these frequencies to the error signal. Specifically, each reference signal $x(n)$ is associated with its NOAS target $y^*(n)$. To determine the near-optimal anti-signal $y^*(n)$, we employ a gradient descent-based algorithm during a pre-processing stage. This stage operates over each example, solving the following optimization problem for each reference signal $x(n)$:

$$\mathbf{y}^* = \arg\min_{\tilde{\mathbf{y}}} \text{NMSE}\left[\mathbf{P} * \mathbf{x}, \mathbf{S} * f_{LS}\{\tilde{\mathbf{y}}\}\right] \tag{11}$$

where $\mathbf{y}^*$ is the near-optimal anti-signal. The optimization starts with a random anti-signal and iteratively adjusts it to minimize the NMSE for the given reference signal $x(n)$. The resulting near-optimal anti-signal $y^*(n)$ is then used to form the target during the fine-tuning stage. To ensure consistency and leverage the prior knowledge gained during the initial training phase, the near-optimal anti-signal, denoted as $\mathbf{y}^*$, is projected onto the anti-signal space associated with the secondary path $S(z)$. This projection plays a crucial role in maintaining the continuity of the training process and is achieved through the use of the secondary path impulse response $\mathbf{S}$. In particular, the near-optimal anti-signal $y^*(n)$ is used to define the following loss function:

$$\mathcal{L}_{\text{NOAS}} = \text{NMSE}\left[\mathbf{S} * f_{LS}\{\mathbf{y}^*\}, \mathbf{S} * f_{LS}\{\mathbf{y}\}\right] \tag{12}$$

Figure 3 illustrates the distinction between optimizing within the canceling signal space versus the anti-signal space. It can be observed that since the NMSE is evaluated at the output of the anti-signal space, optimizing for $\mathbf{S} * \mathbf{y}^*$ facilitates a more straightforward optimization process, given that the starting point $\mathbf{S} * \mathbf{y}$ is closer to the optimal solution $\mathbf{P} * \mathbf{x}$.

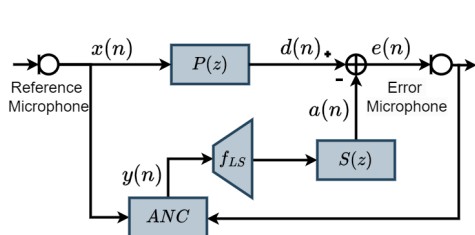

Figure 2: Typical feedforward ANC system diagram.

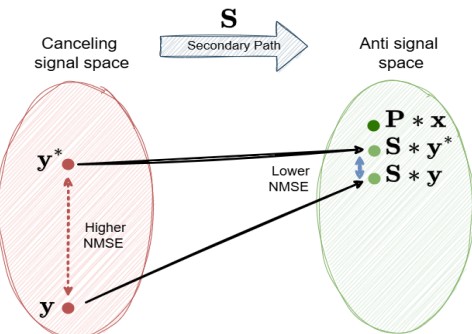

Figure 3: Schematic representation of signal transformations via the secondary path.

## 4 EXPERIMENTS AND RESULTS

**Datasets:** The training data is sourced from the AudioSet dataset (Gemmeke et al., 2017), which we encompassed 248 distinct audio categories. These categories include various types of ambient sounds such as hubbub, speech noise, and speech babble. The dataset comprises 22,224 audio samples, totaling 18.5 hours of audio content. To maintain consistency with ARN (Zhang et al., 2023a), each audio sample was standardized to a duration of 3 seconds and resampled to a 16kHz. Additionally, 20,000 samples (90%) of the dataset were allocated for training. The remaining 2,224 samples were reserved for testing. The test sets were obtained from the NOISEX dataset (Varga & Steeneken, 1993), which includes noisy speech data encompassing a wide range of noise types, such as bubble noise, factory noise, and engine noise. Additionally, we utilized the test sets from the following speech datasets: TIMIT (Garofolo, 1993), which contains recordings from 24 speakers representing 8 dialect regions; LibriSpeech (Panayotov et al., 2015), which includes 40 speakers from audiobook readings; and the Wall Street Journal (WSJ) (Garofolo et al., 1993), which features 8 speakers reading news articles.

**Simulator:** Following previous work (Zhang & Wang, 2021; Zhang et al., 2023a), a rectangular enclosure was modeled to represent the physical setup, with dimensions $[3, 4, 2]$ meters (width, length, height). The room impulse response was generated using the method described by Allen & Berkley (1979). The locations of the microphones and the cancellation load speaker are as follows: the error microphone is located at $[1.5, 3, 1]$ meters, the reference microphone at $[1.5, 1, 1]$ meters, and the cancellation load speaker at $[1.5, 2.5, 1]$ meters. During the training phase, reverberation times were randomly selected from $\{0.15, 0.175, 0.2, 0.225, 0.25\}$ seconds, while in the test phase a reverberation time of 0.2 seconds was used. We utilized the rir_generator package in Python with the high-pass filter option enabled (Allen & Berkley, 1979). The length of the RIR was set to 512 taps. There's a predominant source of nonlinearity stems from the saturation effects inherent in loudspeakers (Ghasemi et al., 2016).

To model the nonlinearity associated with loudspeaker saturation, researchers in the field of ANC commonly (Zhang & Wang, 2021; Zhang et al., 2023a; Mostafavi & Cha, 2023; Cha et al., 2023) employ the Scaled Error Function (SEF), as proposed by Tobias & Seara (2006) $f_{SEF}\{y\} = \int_0^y e^{-\frac{z^2}{2\eta^2}} dz$, where $y$ represents the input to the loudspeaker, while $\eta^2$ quantifies the intensity of the nonlinearity. This function effectively simulates a typical saturation-type nonlinearity, such as the sound level saturation constrained by the physical dimensions of the loudspeaker. The SEF exhibits distinct behaviors at extremes of $\eta^2$: as $\eta^2$ approaches infinity, the function converges to linearity, whereas it approximates a hard limiter as $\eta^2$ tends to zero.

**Hyperparameters:** An extensive grid search and cross-validation were employed to determine the optimal hyperparameters for each method. The hyperparameter values reported here correspond to the configurations that achieved the best performance in our experimental setup. The Deep-AAC architecture was trained using multiple numbers of subbands $Q$, specifically $Q = 0$ (a single full band), 2 and 3. The bands decomposition filters are generated using the $scipy.signal.firwin$ function and applied to the signal via $torch.conv1d$. The temporal duration $M$ was set to 48,000

samples, corresponding to 3-second audio signals sampled at 16 kHz. The channel dimension $C$ was set to 256, and the kernel size $W$ was defined as 16. A batch size of 2 was used for training the DeepAAC architecture. The Adam optimizer (Diederik, 2014) was employed with an initial learning rate of $1.5 \times 10^{-4}$. A learning rate decay factor of 0.5 was applied every 2 epochs after an initial warm-up period of 30 epochs. To mitigate the effects of exploding gradients, gradient clipping was implemented with a threshold of 5.

**Baseline Methods:** We compared our proposed method with several established ANC techniques, including Deep ANC (Zhang & Wang, 2021), Attentive Recurrent Network (ARN) (Zhang et al., 2023a), Filtered-x Least Mean Squares (FxLMS), and Tangent Hyperbolic Function FxLMS (THF-FxLMS, (Ghasemi et al., 2016)). All methods were evaluated in both linear and nonlinear simulations, considering both noise and speech signals. FxLMS, Deep ANC, and ARN were implemented and trained by us. All methods were evaluated under identical simulation conditions. For the learned methods, namely Deep ANC and ARN, the same training dataset used for our proposed method was applied, and we ensured the reproduction of results consistent with those reported in the respective papers. In our Deep ANC implementation, we employed 20-ms short-time Fourier transform (STFT) frames with a 10-ms overlap between consecutive frames. For ARN, we utilized 16-ms frames with an 8-ms overlap. These baseline methods were selected to provide a comprehensive comparison across various ANC paradigms, encompassing both traditional adaptive filtering techniques and more recent deep learning approaches.

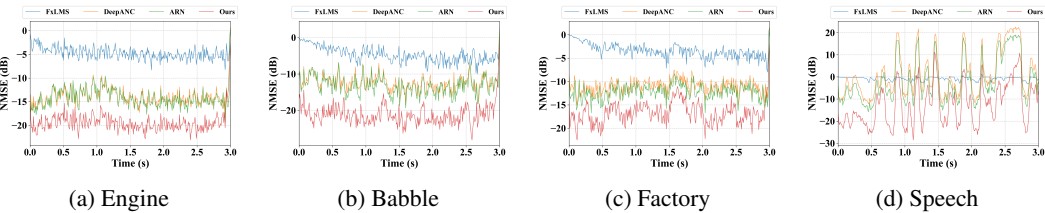

| (a) Engine | (b) Babble | (c) Factory | (d) Speech |

Figure 4: Comparison of NMSE (dB) over time for different noise types.

### 4.1 NOISE CANCELLATION

Table 1 presents the NMSE for ANC algorithms across three noise types—engine, factory, and babble—using 3-second signal segments extracted from the NOISEX-92 dataset. For each noise type, the models were evaluated both without nonlinear distortions (where $\eta^2 = \infty$) and with nonlinear distortions at $\eta^2 = 0.1$ and $\eta^2 = 0.5$. In the case of non-deep learning-based methods, namely FxLMS and THF-FxLMS, gradient clipping at $1e - 4$ was applied due to the sensitivity of these algorithms to the step size, which caused instability during validation. The step sizes for these methods were set to 0.05 for engine noise, 0.4 for factory noise, and 0.3 for babble noise. The results indicate that these algorithms perform suboptimally compared to deep learning-based approaches.

Among the deep learning-based methods, and without considering the nonlinearity saturation effect, the proposed DeepAAC method achieves state-of-the-art results. Specifically, for the case where $\eta^2 = \infty$ it improves performance over the ARN method by 4.29 dB, 4.64 dB, and 7.26 dB for engine, factory, and babble noise, respectively. In the presence of nonlinear distortions ($\eta^2 = 0.5$), DeepAAC continues to outperform ARN, with improvements of 4.36 dB, 4.62 dB, and 7.13 dB for engine, factory, and babble noise, respectively. For more severe nonlinearity ($\eta^2 = 0.1$), DeepAAC still surpasses ARN with gains of 3.79 dB, 4.4 dB, and 5.76 dB. Figures 4a, 4b, and 4c offer visual comparisons of the different methods by plotting NMSE over time. These figures illustrate that the proposed DeepAAC method consistently achieves superior NMSE performance compared to ARN, DeepANC, and FxLMS across almost every time step.

The proposed method was also evaluated for speech enhancement in the presence of noise using active noise cancellation. The PESQ and STOI metrics, presented in Table 3, compare the performance of DeepANC, ARN, and DeepAAC (w/o NOAS) across various SNR levels in the presence of factory noise with nonlinear distortion of $\eta^2 = \infty$. The results demonstrate that DeepAAC outperforms ARN, showing improvements in PESQ scores by 0.7, 0.92, and 0.84 at SNR levels of 5dB, 15dB, and 20dB, respectively. A similar trend is observed for STOI, with enhancements of 0.08, 0.03, and 0.02 for the same SNR levels. Audio samples can be found on the supplementary materials.

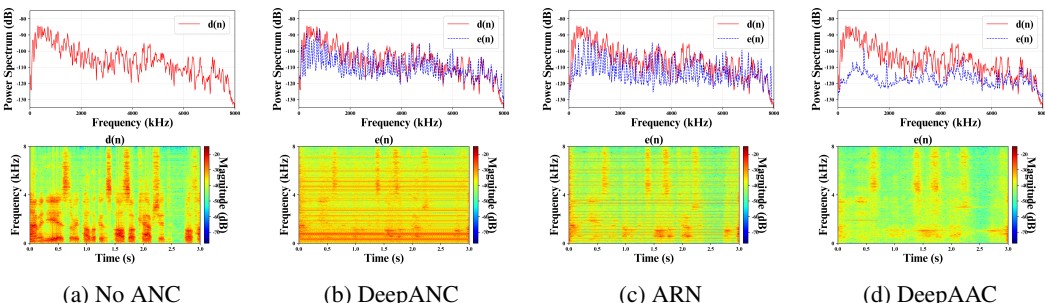

Figure 5: Spectrograms and Power Spectra of Speech Signal (00da010c from WSJ) using Different ANC methods without nonlinear distortions ($\eta^2 = \infty$)

Table 1: Average NMSE ($\downarrow$) in dB for DeepAAC and other algorithms across various noise types and nonlinear distortions. Lower values indicate better performance.

| Method/Noise type | Engine ($\downarrow$) | | | Factory ($\downarrow$) | | | Babble ($\downarrow$) | | |
|---|---|---|---|---|---|---|---|---|---|
| $\eta^2$ | $\infty$ | 0.5 | 0.1 | $\infty$ | 0.5 | 0.1 | $\infty$ | 0.5 | 0.1 |
| FxLMS | -3.38 | -3.33 | -3.32 | -3.27 | -3.17 | -3.11 | -5.39 | -5.33 | -5.30 |
| THF-FxLMS | - | -3.37 | -3.36 | - | -3.26 | -3.24 | - | -5.39 | -5.36 |
| Deep-ANC | -13.96 | -13.91 | -13.6 | -10.7 | -10.69 | -10.62 | -12.42 | -12.4 | -12.22 |
| ARN | -14.59 | -14.59 | -14.38 | -11.61 | -11.61 | -11.54 | -12.91 | -12.9 | -12.72 |
| **DeepAAC** | -18.88 | -18.95 | -18.17 | -16.25 | -16.23 | -15.94 | -20.17 | -20.03 | -18.48 |

## 4.2 SPEECH CANCELLATION

Table 2 presents the average NMSE values for different ANC algorithms across three speech datasets: TIMIT, LibriSpeech, and WSJ, with speech segments affected by varying levels of nonlinear distortions. It is evident that speech cancellation is a more challenging task compared to noise cancellation, as reflected in the performance degradation of the different algorithms.

As observed in the noise cancellation case, in speech cancellation, the non-deep learning methods—FxLMS and THF-FxLMS—demonstrate suboptimal performance compared to deep learning-based approaches. Among the deep learning methods, DeepAAC achieves the best overall results, surpassing the other algorithms significantly.

In the case without nonlinear distortions ($\eta^2 = \infty$), DeepAAC shows improvements over ARN by 6.13 dB, 4.78 dB, and 5.95 dB for the TIMIT, LibriSpeech, and WSJ datasets, respectively. In the presence of moderate nonlinear distortions ($\eta^2 = 0.5$), DeepAAC continues to outperform ARN, with improvements of 6.18 dB for TIMIT, 4.34 dB for LibriSpeech, and 5.99 dB for WSJ. Under more severe nonlinear distortions ($\eta^2 = 0.1$), DeepAAC maintains its superior performance, with enhancements of 5.97dB, 2.46dB, and 5.81dB for TIMIT, LibriSpeech, and WSJ datasets, respectively. Figure 5 illustrates the performance of various ANC methods on a speech signal, comparing power spectra and spectrograms. DeepAAC demonstrates superior noise suppression across all frequencies, including high frequencies, outperforming other methods such as DeepANC and ARN, which struggle more with high-frequency noise. This highlights DeepAAC's effectiveness in providing comprehensive speech cancellation. Figure 4d shows that the property of superior NMSE performance, compared to ARN, DeepANC, and FxLMS at nearly every time step, is also achieved for speech signals. Audio samples can be found on the supplementary materials.

## 4.3 REAL-WORLD SIMULATION

We expanded our investigation to assess the performance of our method in real-world settings, testing it across various simulation scenarios. This was necessary because the fixed task acoustic setup, which relies on the image method, has limitations regarding generalizability and real-world performance. We utilized the dataset from Liebich et al. (2019), which includes acoustic paths from 23

Table 2: Average NMSE ($\downarrow$) in dB for DeepAAC and other algorithms across various speech datasets and nonlinear distortions. Lower values indicate better performance.

| Method/Dataset | TIMIT ($\downarrow$) | | | LibriSpeech ($\downarrow$) | | | WSJ ($\downarrow$) | | |
|---|---|---|---|---|---|---|---|---|---|
| $\eta^2$ | $\infty$ | 0.5 | 0.1 | $\infty$ | 0.5 | 0.1 | $\infty$ | 0.5 | 0.1 |
| FxLMS | -1.39 | -1.36 | -1.26 | -3.43 | -3.40 | -3.28 | -1.92 | -1.90 | -1.85 |
| THF-FxLMS | - | -1.37 | -1.35 | - | -3.41 | -3.39 | - | -1.91 | -1.89 |
| Deep-ANC | -8.52 | -8.56 | -8.48 | -11.92 | -11.81 | -11.08 | -7.54 | -7.55 | -7.51 |
| ARN | -10.31 | -10.27 | -10.2 | -12.87 | -12.74 | -11.87 | -9.48 | -9.48 | -9.42 |
| **DeepAAC** | -16.44 | -16.45 | -16.17 | -17.65 | -17.08 | -14.33 | -15.43 | -15.47 | -15.23 |

Table 3: Average NMSE (dB), STOI and PESQ for deep ANC models in noisy speech situations with LS nonlinearity ($\eta = 0.5$) and factory noise at different SNR levels.

| Method | Noise only | SNR = 5dB | | SNR = 15dB | | SNR = 20dB | |
|---|---|---|---|---|---|---|---|
| | NMSE ($\downarrow$) | STOI ($\uparrow$) | PESQ ($\uparrow$) | STOI ($\uparrow$) | PESQ ($\uparrow$) | STOI ($\uparrow$) | PESQ ($\uparrow$) |
| Deep-ANC | -10.69 | 0.83 | 1.39 | 0.93 | 2.10 | 0.96 | 2.45 |
| ARN | -11.61 | 0.84 | 1.51 | 0.94 | 2.43 | 0.96 | 2.92 |
| DeepAAC | -15.94 | 0.92 | 2.21 | 0.97 | 3.35 | 0.98 | 3.76 |

individuals, measured in the real world and encompassing both primary and secondary paths. We applied DeepAAC, along with baseline approaches, to the updated simulation conditions and assessed their performance using Factory and Babble noise from the NoiseX-92 dataset, in addition to speech samples from the WSJ dataset. The results in Table 6 present the average NMSE across these categories. The results demonstrate that DeepAAC consistently outperforms the alternative methods, achieving improvements of 2.80dB in the Factory noise, 2.70dB in the Babble noise, and 1.53dB on the WSJ dataset.

## 4.4 MODEL ANALYSIS

In the DeepAAC architecture, the number of frequency bands is a crucial hyperparameter that directly influences performance. Table 4 presents a comparative analysis of the performance of DeepAAC across different band configurations for the Factory noise, TIMIT, LibriSpeech, and WSJ datasets, with the nonlinearity factor set to $\eta^2 = 0.5$. The "1-band" configuration corresponds to a full single-band model, whereas the "3-band" configuration comprises one full medium band and two smaller sub-bands. Similarly, the "4-band" configuration includes one full medium band along with three smaller sub-bands. Due to computational resource constraints and the increased model complexity—particularly with the 4-band configuration, which requires 40M parameters—further configurations were not evaluated. It is important to note that a 2-band architecture was not considered, as in the DeepAAC framework, it would consist of two full bands, which was not the intended design.

As illustrated in Table 4, increasing the number of bands leads to an overall improvement in model performance. For instance, the 4-band configuration outperforms the 3-band variation by $0.58$ dB, $0.19$ dB, $0.37$ dB, and $0.48$ dB for the Factory noise, TIMIT, LibriSpeech, and WSJ datasets, respectively. This improvement is attributed to the model's enhanced ability to focus on specific sub-frequency bands, which is particularly advantageous for handling higher frequency components in speech. Table 5 presents a comparison of model size and performance, where the NMSE is evaluated on factory noise with nonlinear distortion of $\eta = 0.5$. All DeepAAC variants in this comparison are without NOAS optimization. The results indicate that even the smallest DeepAAC configuration (1-band, small) surpasses the ARN architecture by $1.85$ dB, despite utilizing half the number of parameters (8.0M versus 15.9M). This is a significant outcome given the critical importance of model size in real-time active noise cancellation (ANC) scenarios, where latency constraints play a pivotal role. Additionally, the 3-band configuration achieves superior results compared to a single large-band variant, despite the latter having 3.1M more parameters, underscoring the critical role of the multi-band approach in enhancing performance.

Table 4: Average NMSE ($\downarrow$) in dB of our method (**w/o NOAS**) for Noise and Speech using different number of bands, with nonlinear distortion of $\eta^2 = 0.5$.

| Method/Dataset | #Bands | Factory ($\downarrow$) | TIMIT ($\downarrow$) | LibriSpeech ($\downarrow$) | WSJ ($\downarrow$) |
|---|---|---|---|---|---|
| DeepAAC (small) | 1 | -13.46 | -14.26 | -14.88 | -13.22 |
| DeepAAC (medium) | 1 | -15.19 | -15.82 | -16.56 | -14.86 |
| DeepAAC | 3 | -15.94 | -16.36 | -16.95 | -15.32 |
| DeepAAC | 4 | -16.52 | -16.55 | -17.41 | -15.84 |

The computational complexity of the models was assessed by comparing their FLOPs, averaged across 20 three-second samples from the Noisex-92 dataset, as presented in Table 7. The single-band, small variant of DeepAAC demonstrated exceptional efficiency, requiring only 2.862G FLOPs while consistently surpassing the performance of the other models. This highlights its superior balance between computational cost and effectiveness. Table 8 demonstrates the superiority of optimizing within the anti-signal space, as it yields improved performance. Specifically, the NMSE distance between $\mathbf{y}$ and $\mathbf{y}^*$ is notably greater than the distance between $\mathbf{P} * \mathbf{x}$ and $\mathbf{S} * \mathbf{y}$, highlighting the effectiveness of this approach. Additionally, it is evident that the use of the NOAS optimization approach (middle column) yields an improvement of $1.07$ dB, further validating the superiority of this method.

## 5 CONCLUSION

In this paper, we introduced a novel AAC approach using the Multi-Band Mamba architecture, advancing deep learning-based noise and audio 'cancellation. By partitioning audio into frequency bands, our method enhances anti-signal generation and phase alignment. Combined with an optimization-driven loss function, it achieves near-optimal performance, improving both ANC and AAC outcomes. Our experimental results demonstrate a significant performance boost compared to state-of-the-art baselines, with improvements of 7.2dB in ANC and 6.2dB in AAC for voice audio signals.

These results confirm the multi-band architecture's effectiveness in handling diverse frequencies and real-world acoustic environments, where traditional methods often fail. Our approach addresses key challenges in the field by effectively leveraging frequency decomposition and optimization-based anti-signal generation, paving the way for more advanced audio cancellation technologies.

Table 5: Comparison of different deep learning based ANC methods based on parameter size.

| Models | #Params | NMSE ($\downarrow$) |
|---|---|---|
| Deep-ANC | 8.8M | -10.69 |
| ARN | 15.9M | -11.61 |
| DeepAAC, 1 Band, S | 8.0M | -13.46 |
| DeepAAC, 1 Band, M | 15.8M | -15.19 |
| DeepAAC, 1 Band, L | 34.0M | -15.72 |
| DeepAAC, 3 Bands | 31.9M | -15.94 |
| DeepAAC, 4 Bands | 40.0M | -16.52 |

Table 6: Average NMSE ($\downarrow$) in dB for different deep learning based ANC methods on noise and speech signals, evaluated on real-world measured $P$ and $S$ with a nonlinearity term of $\eta^2 = 0.5$.

| Method/Dataset | Factory | Babble | WSJ |
|---|---|---|---|
| DeepANC | -9.29 | -10.94 | -8.26 |
| ARN | -8.97 | -11.17 | -10.70 |
| Ours | -12.09 | -13.87 | -12.23 |

Table 7: FLOPs and NMSE comparison for different deep learning based ANC methods.

| Method | FLOPs (G) ($\downarrow$) | NMSE ($\downarrow$) |
|---|---|---|
| DeepANC | 7.199 | -10.69 |
| ARN | 5.281 | -11.61 |
| Ours | 2.419 | -13.46 |

Table 8: Comparison of NMSE ($\downarrow$) distances for different objectives, with and without NOAS optimization.

| Method | $[\mathbf{y}^*, \mathbf{y}]$ | $[\mathbf{P} * \mathbf{x}, \mathbf{S} * \mathbf{y}]$ | $[\mathbf{S} * \mathbf{y}^*, \mathbf{S} * \mathbf{y}]$ |
|---|---|---|---|
| - NOAS | -9.85 | -16.53 | -18.56 |
| + NOAS | -12.77 | -17.60 | -19.62 |

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

# A APPENDIX

## A.1 LIMITATION AND FUTURE WORK

The Multi-Band Mamba architecture demonstrates significant effectiveness in AAC. However, it is not without limitations that warrant further investigation. A key drawback lies in the trade-off between performance and complexity. While our approach achieves enhanced cancellation across a wide frequency spectrum, the increased model complexity associated with handling multiple frequency bands results in higher computational costs. This limitation renders the method less practical for low-latency applications or devices with restricted processing capabilities.

To address this limitation, future work should focus on reducing computational overhead and enabling real-time processing. This could involve exploring dynamic or adaptive frequency band partitioning strategies that tailor the model's complexity to the characteristics of the input signal.

## A.2 ABLATION STUDY

To evaluate the contributions of the principal components of the DeepAAC architecture, an ablation study was conducted. This study focused on four critical aspects: multiband processing, the influence of band size (small vs. medium), the impact of NOAS optimization, and the effect of the Mamba architecture.

The analysis results concerning multiband processing, band size, and NOAS optimization are detailed in Table 9 , which reports the NMSE performance across four distinct datasets: Factory, TIMIT, LibriSpeech, and WSJ, all evaluated under nonlinear distortion conditions ($\eta = 0.5$). In our notation, "+ S - Multiband - NOAS" refers to a small band configuration (8 mamba layers) without multiband processing or NOAS optimization, while "+ S - Multiband + NOAS" refers to the same small band architecture with NOAS optimization applied. Similarly, "+ M - Multiband - NOAS" represents a medium band configuration (16 mamba layers) without NOAS, and "+ M - Multiband + NOAS" applies NOAS optimization to the same medium band model. The **Full Method** is defined as a configuration that employs one full medium band and two small sub-bands, with NOAS optimization applied. All models were initially trained using the ANC loss function defined in Eq. 10. Configurations with "+ NOAS" were fine-tuned using NOAS optimization, whereas configurations with "- NOAS" were trained exclusively using the ANC loss in Eq. 10. The results demonstrate that the removal of NOAS optimization consistently degrades performance across all datasets. For instance, on the Factory dataset, applying NOAS optimization to the small band model leads to a

Table 9: Average NMSE (↓) in dB for noise and speech using multiple variants of DeepAAC, with nonlinear distortion of $\eta = 0.5$.

| Method/Dataset | Factory (↓) | TIMIT (↓) | LibriSpeech (↓) | WSJ (↓) |
|---|---|---|---|---|
| + S - MultiBand - NOAS | -13.46 | -14.26 | -14.88 | -13.20 |
| + S - MultiBand + NOAS | -14.19 | -14.54 | -15.24 | -13.55 |
| + M - MultiBand - NOAS | -15.19 | -15.82 | -16.56 | -14.86 |
| + M - MultiBand + NOAS | -16.09 | -16.25 | -16.92 | -15.27 |
| **Full Method** | -16.23 | -16.45 | -17.08 | -15.47 |

Table 10: Average NMSE (↓) in dB for different deep-learning architectures on multiple datasets with nonlinearity term of $\eta^2 = 0.5$. Numbers in parentheses indicate the parameter count for each model.

| Method/Dataset | Factory (↓) | TIMIT (↓) | WSJ (↓) | Librispeech (↓) |
|---|---|---|---|---|
| Convolution (41.9M) | -4.62 | -6.57 | -6.43 | -6.80 |
| LSTM (37.5M) | -12.17 | -11.83 | -11.88 | -12.99 |
| Transformer (34M) | -12.60 | -12.90 | -12.04 | -13.86 |
| Ours (31.9M) | -15.94 | -16.36 | -15.32 | -16.95 |

performance improvement of 0.73dB, while the medium band model shows a larger improvement of 0.90dB. This trend holds across the other datasets, reinforcing the crucial role of NOAS optimization in enhancing model performance. Multiband processing further improves the overall effectiveness of DeepAAC. For example, the **Full Method** consistently outperforms the "+ M - Multiband + NOAS" configuration, with gains of 0.14dB, 0.2dB, 0.16dB, and 0.2dB on the Factory, TIMIT, LibriSpeech, and WSJ datasets, respectively. We will discuss multiband processing importance further in the next section. Interestingly, the performance of the "+ S - Multiband - NOAS" configuration is consistently lower than that of the "+ M - Multiband - NOAS" variant across all datasets. Specifically, the small band model underperforms by 1.73dB on Factory, 1.56dB on TIMIT, 1.68dB on LibriSpeech, and 1.66dB on WSJ. This indicates that while multiband processing is valuable, the choice of band size plays a significant role in the model's performance, with larger band sizes, particularly when combined with NOAS, yielding the best results.

We evaluated the impact of the Mamba block by comparing its performance to Transformers, LSTMs, and CNNs, as shown in Table 10. We utilized a three-band configuration comprising two small sub-bands and one medium-sized band. For fairness, DeepAAC's core modules ($E_0, \ldots, E_Q$ and $D$) were retained as originally designed. For the Transformer baseline, we utilized an ARN-based model with $d_{model} = 512$, a single layer for the sub-band processing, and two layers for the full-band processing. In the LSTM setup, we used $torch.LSTM$ with two layers for the sub-band processing and four for the full-band processing (hidden size = 256). For the convolutional baseline, we adapted a convolutional autoencoder architecture derived from the DeepANC skeleton, omitting the LSTM components, with four encoder layers and four decoder layers with batch normalization applied after each layer. Although this configuration is effective in capturing local features, it reflects a relatively basic convolutional neural network (CNN) architecture. As such, it does not incorporate the more recent innovations in CNN design, which could explain the suboptimal performance observed in our results. The kernel size for the signal sub-bands was set to $1 \times 2 \times 2$, while for the full-band signal, the kernel size was $1 \times 2 \times 4$, with the final dimension denoting the kernel depth. The Mamba architecture achieved substantial gains, surpassing Transformers by 3.34 dB in Factory noise and by 3.66, 5.28, and 3.09 dB on TIMIT, WSJ, and Librispeech datasets, respectively. These results emphasize the Mamba block's effectiveness and its value in the DeepAAC framework for robust active noise cancellation.

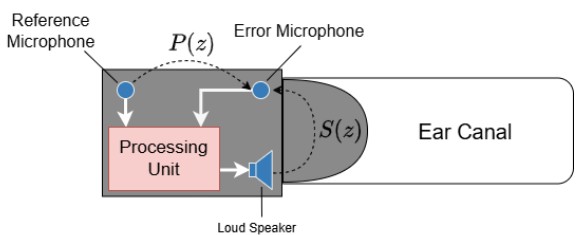

Figure 6: Schematic representation of the acoustic paths in an in-ear ANC system.

### A.3 ACOUSTIC CONDITIONS VISUALIZATION

To elucidate the acoustic dynamics of ANC systems in in-ear headphones, Figure 6 presents a schematic representation of the Primary and Secondary acoustic paths.

The primary path $P(z)$ characterizes the transfer function between the external noise, as captured by the reference microphone, and the error microphone. This path models the propagation of ambient noise through the system. Conversely, the secondary path $S(z)$ represents the transfer function from the loudspeaker to the error microphone, encompassing the acoustic feedback loop within the ear canal.

The schematic highlights the interaction between critical system components, including the processing unit, reference microphone, error microphone, and loudspeaker.

