# OpenReview forum: "Active Audio Cancellation with Multi-Band Mamba Network"
_ICLR.cc/2025/Conference — ICLR 2025 Conference Withdrawn Submission_

### Official Review · Reviewer_mDwo · 2024-10-31

**Soundness:** 3
**Presentation:** 3
**Contribution:** 3
**Rating:** 8
**Confidence:** 4

**Summary:**

This paper proposes a Active Noise Cancellation (ANC) network. It represents the first attempt to actively cancel general audio signals with deep learning. The paper introduces a segmentation strategy for separate modeling and incorporates Mamba as a sequence modeling module. Additionally, a new training strategy is proposed during training to generate more fitting anti-signals.

**Strengths:**

This paper is the first to consider actively canceling general audio signals to address the issues inherent in Active Noise Cancellation. It incorporates a frequency band segmentation strategy in its design to better focus on the characteristics of each frequency band and introduces Mamba for sequence modeling. In addition to structural innovations, the paper proposes a Near Optimal Anti-Signal Optimization strategy to solve the problem of different paths producing different frequencies.

**Weaknesses:**

The first issue is an expression problem at the beginning of Section 4.1, which should state "Table 1 presents the NMSE for ANC algorithms..." instead of "Table 4 presents the NMSE for ANC algorithms...". The second issue is that the baseline models used for comparison are relatively old and do not strongly demonstrate that this model has reached the state-of-the-art (SOTA). The third issue is that the proposed optimization strategy first finds y' based on the given reference signal x, and then finds the final target y based on y'. Essentially, it is still based on the reference signal for the loss function. Therefore, the significance of this approach and why it leads to performance improvements need to be clarified.

**Questions:**

see in weaknesses

---

> ### Author Response · Authors · 2024-11-22
> **Addressing Review Comments**
>
> We thank the reviewer for the valuable feedback.
>
> **Weaknesses:**
> \
> **W1: The first issue is an expression problem at the beginning of Section 4.1, which should state "Table 1 presents the NMSE for ANC algorithms..." instead of "Table 4 presents the NMSE for ANC algorithms...".**
> \
> **A1:** Thank you for pointing this out. You are correct, and we will fix the expression in the revised version to state "Table 1 presents the NMSE for ANC algorithms" instead of "Table 4."
>
> \
> **W2:The second issue is that the baseline models used for comparison are relatively old and do not strongly demonstrate that this model has reached the state-of-the-art (SOTA).**
> \
> **A2:** Thank you for your feedback. If we missed any relevant recent baselines, please let us know, and we will gladly include them in our comparison. For reference, we did include ARN, which was published in 2023, as one of the baselines in our evaluation.
>
> \
> **W3: The third issue is that the proposed optimization strategy first finds y' based on the given reference signal x, and then finds the final target y based on y'. Essentially, it is still based on the reference signal for the loss function. Therefore, the significance of this approach and why it leads to performance improvements need to be clarified.**
> \
> **A3:** Thank you for raising this point. The significance of our approach lies in addressing a potential issue where the secondary path filter (S) attenuates certain frequencies that the primary path filter (P) does not. In such cases, during training, the model could be penalized for high energy at the error microphone in these frequencies, even though it cannot realistically resolve this discrepancy.
> To avoid such undue penalties, we optimize over y′ using the NOAS optimization method. This ensures that the secondary path (S) is present on both sides of the loss function, effectively nullifying these problematic cases. This optimization strategy allows the model to focus on achievable improvements and contributes to the observed performance gains.
> We will clarify this in the revised version of the paper.

---

### Official Review · Reviewer_m3g7 · 2024-10-31

**Soundness:** 3
**Presentation:** 2
**Contribution:** 3
**Rating:** 6
**Confidence:** 5

**Summary:**

Noise cancellation remains an important problem, even if it has been studied for a long time and machine learning can be a useful component. This paper addresses a particular subcategory of noise cancellation, active audio cancellation, where, apparently, no prior machine learning models have been proposed. While noise cancellation is important, I did not find the subcategory particularly important, and there were no supporting arguments for this application scenario.
The methodology used in the study is mostly high quality. My only worry is that the fixed physical configuration of microphones and loudspeaker will diminish the practical value of the results.

**Strengths:**

The background is explained in admirable detail.
The machine learning methodology is high quality.
Writing is good.

**Weaknesses:**

The application scenario is not explained sufficiently to motivate the *audio* cancellation.
Fixed physical setup of sound sources and sinks limits the generality of results.

**Questions:**

Problem statement
- The differentiation between *noise* cancellation and not active *audio* cancellation is questionable. For typical applications, any sound that is not desired is noise, and the undesired sounds can be any type of audio. The authors are probably right in that noise cancellation has been applied for most application scenarios, but that is the typical desired functionality. It is useful to emphasise the most typical scenarios in training; thus, focusing on noise is usually beneficial. Authors should, therefore, demonstrate why active *audio* cancellation is a desired functionality. What is the expected application scenario where generic audio is the expected distortion?
The vocabulary is unclear; how do you define the secondary path? A figure would have been helpful in Section 3.1. It might be that Figure 2 was intended for this purpose, but it was never referenced in the text. However, that figure does not show how sounds travel in the acoustic space.

Experiment setup
- The setup is otherwise good, but it can be a problem that the physical configuration of microphones and loudspeakers were fixed. This implies that the model would have to be trained for every particular scenario, which is impractical. This is a potential problem unless experiments demonstrate otherwise, but this was not part of the experiments. At least, it has to be mentioned as a drawback of the proposed method.

Minor comments
- In Eq 4, for visual clarity, please scale the parenthesis to match the content inside them (in LaTeX this can be done by \left and \right, before the parenthesis sign)
- Please mention the method you use for frequency decomposition

---

> ### Author Response · Authors · 2024-11-22
> **Addressing Review Comments #1**
>
> We acknowledge the reviewer's observations.
>
> \
> **Weaknesses:**
> \
> **W1: The application scenario is not explained sufficiently to motivate the audio cancellation. Fixed physical setup of sound sources and sinks limits the generality of results.**
> \
> **A1:** This setup aligns with the standard task definition in the field, as established in foundational works like in [1-4]. Our work builds on this well-defined scenario by advancing the ability to cancel speech, not just typical noises, which has been a less-explored challenge. Extending to more generalized setups is indeed an exciting avenue for future work.
>
> \
> **Questions:**
> \
> **Q1: Problem statement: The differentiation between noise cancellation and not active audio cancellation is questionable. For typical applications, any sound that is not desired is noise, and the undesired sounds can be any type of audio.**
> \
> **A1:** We appreciate the reviewer’s insightful comment and apologize for the lack of clarity in our terminology. In this work, “audio cancellation” encompasses both typical noise and speech signals. While most ANC systems indeed focus on general noise suppression, our study addresses an additional challenge: the cancellation of speech signals, which are notably harder to suppress, as demonstrated by the results in Table 2 (speech signals) compared to Table 1 (typical noises).  We will clarify this distinction in the final version of the paper.
>
> \
> **Q2: The authors are probably right in that noise cancellation has been applied for most application scenarios, but that is the typical desired functionality. It is useful to emphasise the most typical scenarios in training; thus, focusing on noise is usually beneficial. Authors should, therefore, demonstrate why active audio cancellation is a desired functionality. What is the expected application scenario where generic audio is the expected distortion?**
> \
> **A2:** Thank you for the helpful feedback. We agree that noise cancellation is the typical desired functionality in most applications, but our work focuses on addressing the case where speech itself is the undesired sound. This is particularly relevant in scenarios such as private environments or shared spaces where speech leakage can disrupt privacy or create disturbances.
>
> \
> **Q3: The vocabulary is unclear; how do you define the secondary path?**
> \
> **A3:** Thank you for pointing that out. The "secondary path" refers to the acoustic path between the loudspeaker and the error microphone. We apologize for the lack of clarity and will make sure to define this term more clearly in the revised version of the paper.
>
> \
> **Q4: A figure would have been helpful in Section 3.1. It might be that Figure 2 was intended for this purpose, but it was never referenced in the text. However, that figure does not show how sounds travel in the acoustic space.**
> \
> **A4:** We will fix the issues you've raised. While we did reference Figure 2 at the end of Section 3.1, we understand that the figure does not clearly demonstrate how sounds travel in the acoustic space. To address this, we will add an additional figure to explicitly show the signal’s path in the acoustic space, and we will ensure proper referencing of all figures throughout the text.
>
> \
> **Q5: Experiment setup: The setup is otherwise good, but it can be a problem that the physical configuration of microphones and loudspeakers were fixed. This implies that the model would have to be trained for every particular scenario, which is impractical. This is a potential problem unless experiments demonstrate otherwise, but this was not part of the experiments. At least, it has to be mentioned as a drawback of the proposed method.**
> \
> **A5:** We appreciate the reviewer’s feedback. The experimental setup, where the physical configuration of microphones and loudspeakers is fixed, aligns with standard task definitions in the field, as demonstrated by foundational works like [1-4]. While we acknowledge that fixed configurations could limit the generalizability of the method in real-world scenarios, we believe that extending the framework to more dynamic setups is an exciting avenue for future research. We will clarify this limitation in the revised manuscript and include it as a drawback of the proposed method.
>
> \
> **Q6: Minor comments: In Eq 4, for visual clarity, please scale the parenthesis to match the content inside them (in LaTeX this can be done by \left and \right, before the parenthesis sign)**
> \
> **A6:** Thank you for the suggestion. We will modify Eq. 4 to use the \left and \right commands in LaTeX to ensure the parentheses match the content inside.

---

> > ### Author Response · Authors · 2024-11-22
> > **Addressing Review Comments #2**
> >
> > **Q7: Minor comments: Please mention the method you use for frequency decomposition**
> > \
> > **A7:** Our frequency decomposition method uses one full-band filter that passes the entire signal and additional sub-band filters, each covering a frequency range of equal size. Specifically, if there are QQQ sub-bands and the highest frequency is FFF, the iii-th sub-band filter covers the frequency range (i-1)F/Q to iF/Q,  where i=1,2,…,Q.
> > We generate these filters using the scipy.signal.firwin function.
> > Once generated, we apply the filters to the signal using torch.conv1d
> > We will add these details to the final version of the paper for clarity.
> >
> >
> > \
> > **Refrences:**
> > \
> > [1] Deep ANC: A deep learning approach to active noise control
> > \
> > [2] Low-Latency Active Noise Control Using Attentive Recurrent Network
> > \
> > [3] Deep learning-based active noise control on construction sites
> > \
> > [4] DNoiseNet: Deep learning-based feedback active noise contro

---

> > ### Comment · Reviewer_m3g7 · 2024-11-22
> > **Most of my comments were dismissed by "this is what everyone else does"**
> >
> > My comments regarding audio vs noise cancellation were dismissed with the argument that this is how the community usually approaches the issue. The logical argument has the same structure as the argument: if the community chooses to jump off a cliff, so would you.
> > I strongly believe that you, as authors, are responsible for the contents of your paper and cannot offload that to prior work. Using the same methodology and metrics is, however, useful since it allows comparison with prior work. Irrespective of that, you should always *also* use the appropriate methods and metrics and mention solid motivations, so that you do not propagate bad habits.

---

> > > ### Author Response · Authors · 2024-11-26
> > > **Addressing Review Comments #3**
> > >
> > > We thank the reviewer for their thoughtful feedback. Our response has two parts: first, addressing the concern about fixed physical setups, and second, clarifying the differentiation between noise cancellation and audio cancellation.
> > >
> > > >The setup is otherwise good, but it can be a problem that the physical configuration of microphones and loudspeakers were fixed. This implies that the model would have to be trained for every particular scenario, which is impractical. This is a potential problem unless experiments demonstrate otherwise, but this was not part of the experiments. At least, it has to be mentioned as a drawback of the proposed method.
> > >
> > > \
> > > Thank you for your valuable feedback. Our research acknowledges the inherent challenges and practical limitations of fixed physical configurations in ANC systems. Indeed, a recurring theme in the ANC literature is the critical importance of precise acoustic path knowledge for effective and robust performance. This is emphasized, for example, in Fabry et al., 2020 [1] which highlights the necessity of tailored secondary path measurements and the constraints of runtime primary path estimation. Similarly, works like Fabry et al., 2019 [2] stress that accurate transfer function measurements, often requiring controlled environments, are pivotal for achieving high performance, even while recognizing the impracticality of some runtime measurements.
> > > These works emphasize that robust ANC performance necessitates accurate path knowledge, and runtime measurement of paths—particularly the primary path—is highly impractical without external hardware or controlled environments.
> > > \
> > > We find your concern regarding the generalizability of our scenario particularly insightful. It is essential to clarify that while our study utilized a specific physical configuration, this setup is not an end in itself but a foundational step. To address the generalizability concern, we extended our evaluation using diverse datasets, notably Liebich et al., 2019 [3]. This dataset includes various real-world conditions, such as transfer functions measured for 23 participants across different scenarios. The results of our model on these paths, summarized below, demonstrated its robustness and adaptability. The value is the average NMSE (dB) across Factory, Babble (from NoiseX-92) and WSJ categories.
> > > |         | ARN   | DeepANC | Ours  |
> > > |---------|-------|---------|-------|
> > > | Factory | -8.97  | -9.29    | -12.09 |
> > > | Babble  | -11.17 | -10.94   | -13.87 |
> > > | WSJ     | -10.70 | -8.26    | -12.23 |
> > >
> > >
> > > This provides empirical evidence that our approach can generalize beyond the fixed configuration initially described.
> > > Furthermore, since ANC systems are often designed for specific environments or use cases (e.g., headphones, enclosures), it is standard practice to evaluate under controlled conditions to establish baseline performance.
> > >
> > > \
> > > We acknowledge that fully dynamic configurations are an important area of future exploration. On the path toward this ultimate goal, our current findings contribute meaningfully to the development of robust ANC systems and demonstrate adaptability to diverse acoustic paths.
> > >
> > > \
> > > **References:**
> > > \
> > > [1] Primary Path Estimator Based on Individual Secondary Path for ANC Headphones
> > > \
> > > [2] Acoustic Equalization for Headphones Using a Fixed Feed-Forward Filter
> > > \
> > > [3] Acoustic Path Database for ANC In-Ear Development

---

> ### Author Response · Authors · 2024-11-26
> **Addressing Review Comments #4**
>
> >My comments regarding audio vs noise cancellation were dismissed with the argument that this is how the community usually approaches the issue. The logical argument has the same structure as the argument: if the community chooses to jump off a cliff, so would you. I strongly believe that you, as authors, are responsible for the contents of your paper and cannot offload that to prior work. Using the same methodology and metrics is, however, useful since it allows comparison with prior work. Irrespective of that, you should always also use the appropriate methods and metrics and mention solid motivations, so that you do not propagate bad habits.
>
> \
> Thank you for your feedback. We appreciate your concern about the clarity and rigor of our methodology and metrics, as well as the need to prevent the propagation of suboptimal practices in the field.
> In our paper, we use the term "audio cancellation" to distinguish our task from the more commonly studied "noise cancellation." This differentiation is intentional, as our work addresses not only the removal of environmental noise but also the cancellation of speech, which has distinct and meaningful applications.
> While noise cancellation remains critical for various applications, speech cancellation is an emerging need in specific scenarios, such as:
> \
> **Public and shared spaces:** For instance, reducing speech distractions in open offices, libraries, buses, or restaurants.
> \
> **Virtual Reality (VR) and Augmented Reality (AR):** Ensuring users hear only virtual sounds by filtering out external voices.
> \
> **Educational settings:** Enabling a classroom environment where students focus on the teacher's voice, free from other ambient conversations.
> \
> **Call centers and customer support:** Enhancing clarity by isolating the agent’s voice from other speakers.
>
> \
> We would like to clarify and expand on our reasoning for following the prevailing community standards while ensuring we do not propagate bad practices.
> Indeed, the statistical properties of speech and noise differ significantly, which is a key consideration in our approach. Speech typically occupies a broad spectrum of frequencies, often with a higher frequency range due to the presence of harmonics, compared to many common noise sources that exhibit more concentrated or lower-frequency energy distributions.
> This disparity directly impacts the performance of ANC algorithms, as demonstrated in our results. Specifically, Table 1 (noise cancellation) and Table 2 (speech cancellation) in our paper illustrate the distinct performance degradation across algorithms when transitioning from noise to speech cancellation scenarios.
> In this context, DeepAAC's superior performance across all frequency bands, including high frequencies, further validates our methodological choices (as shown in Figure 5 in the paper), This ability to handle high-frequency components highlights DeepAAC's adaptability and underscores the importance of addressing these spectral differences explicitly.

---

### Official Review · Reviewer_5EHs · 2024-11-02

**Soundness:** 2
**Presentation:** 2
**Contribution:** 2
**Rating:** 3
**Confidence:** 4

**Summary:**

This paper presents a deep learning-based approach for active audio cancellation (AAC), which is a broader problem that encompasses active noise cancellation (ANC) as a special case, where the goal is to suppress general audio signals rather than only the typical noise-like signals. To better handle the more complex case of general audio than just cancellation of noise, the authors present DeepAAC, a deep neural network model featuring three key components: i) multi-band processing, ii) the Mamba architecture (a recent state space model (SSM)), and iii) a new loss function utilizing the so-called "near optimal anti-signa." Experimental results on simulated audio data show the advantages of the proposed method over conventional adaptive filtering-based approaches as well as recent deep learning-based models in terms of the normalized mean squared error (NMSE) across several speech and noise types and settings of loudspeaker distortion. Speech quality evaluations are also provided through speech enhancement experiments in addition to NMSE measurements.

**Strengths:**

- Utilizing the state space model (Mamba) in the context of active audio or noise cancellation seems novel.

- Comparison with both conventional adaptive filtering approaches (least mean square (LMS) type algorithms) and deep learning-based models is nice.

- Demonstrating better performance over existing approaches across several speech and noise types, including consideration of different settings of the loudspeaker nonlinearity, and various signal-to-noise ratios (SNRs) in speech enhancement tasks.

**Weaknesses:**

- The proposed components lack evidence to well support their respective merits. To be more specific,
 > - **Multi-band**: Although Table 4 presents improved NMSE as the number of bands increases, it is not clear if such improvement is obtained due to using multiple frequency bands, or just mainly due to having a larger model size for the case of more bands (e.g., 1 band (15.8M) vs. 4 bands (40M) given in Table 6). Therefore, it is suggested that the authors compare multi-band models to single-band models with equivalent or similar parameter counts, to more conclusively demonstrate the benefits of the multi-band approach.
> - **Mamba blocks**: The benefits brought by using the SSM (Mamba) layers are not demonstrated. It would be nice if the authors conduct ablation studies comparing Mamba to other architectures like transformers, LSTMs, or CNNs while keeping other aspects of the model constant. This would provide clearer evidence of Mamba's advantages in this context.
> - **NOAS loss**: The motivation of using the near optimal anti-signal (NOAS) loss in eq. (12) is not clear. On lines 239-243, it states that `` The difficulty arises because if the secondary path $S(z)$ attenuates certain frequencies, the error signal $e(n)$ in these bands will be non-zero since the primary signal $d(n)$ may contain energy in these frequencies. Consequently, the ANC controller will encounter errors during the training process regardless of its estimation of the anti-signal $y(n)$.'' However, as shown in eq. (11), the secondary path $S$ is still presented in the optimization criterion. As a result, the aforementioned difficulty due to the attenuation effect of $S(z)$ still remains. Then, how does the NOAS help in alleviating such issues? Please provide a more detailed explanation or mathematical justification for how the NOAS loss addresses the issues with frequency attenuation in the secondary path. Additional comparative experiments showing the performance with and without the NOAS loss on such aspects could also help clarify its benefits.

- The other weakness of the paper is that the proposed DeepAAC is evaluated on simulated data using a room acoustics simulator. It is more desirable to also test the system on real-world recorded audio data with an actual headset, or at least using real-world measured acoustic paths $P(z)$ and $S(z)$ to simulate the test samples (e.g., *Liebich, Stefan, et al. "Acoustic path database for ANC in-ear headphone development," Universitätsbibliothek der RWTH Aachen, 2019*), for better demonstrating the proposed method's capabilities.

**Questions:**

- In Section 3.4, could you elaborate on the gradient descent-based algorithm employed for eq. (11) during a pre-processing stage? Also, as there is no nonlinear terms in eq. (11), can the optimal solution $\mathbf{y}^*$ be given by just solving the linear least squares problem (where a closed-form solution is available) of it? If yes, could you clarify why choosing a gradient descent-based algorithm over a closed-form solution, and what advantages, if any, this iterative approach offers over direct linear least squares solving?

- Should the $\mathbf{s}$ in Figure 3 be upper-case $\mathbf{S}$?

- The font size and resolution of plots in Figure 4 and Figure 5 can be improved for better viewing.

-  Section 4.1, line 347, should it be "Table 1 presents.." instead of "Table 4 presents..."?

- For ablation studies in Section 4.3, does ``without NOAS optimization'' mean just using the $\mathcal{L}_{\text{ANC}}$ in eq. (10)? Please explicitly state what loss function is used when NOAS optimization is not applied.

- In Section 4.3, lines 427-428, why is it interesting to see that the "+S-Multiband-NOAS" configuration performance is lower than "+M-Multiband-NOAS"? The former has a much smaller model size (8M) according to Table 6 than the latter (15.8M), so it is not surprising that the performance is worse of the former.

---

> ### Author Response · Authors · 2024-11-22
> **Addressing Review Comments #1**
>
> Thank you for your insightful comments and suggestions.
>
> \
> **Weaknesses:**
> \
> **W1: Multi-band:**
>
> We understand the concern regarding the potential impact of model size on the performance improvement observed with the multi-band approach, as compared to a single-band model. To address this, we conducted an additional experiment with a large single-band model, using 34M parameters (as compared to the 31.9M parameters for the multi-band architecture with 3 bands).
> The table below presents the average NMSE results from this experiment, with all models incorporating a nonlinearity term of η=0.5. The numbers in parentheses indicate the parameter count for each model.
>
> | Dataset         | Single-Band (34M) | Multi-Band (31.9M) |
> |-----------------|----------------------------|------------------|
> | Factory         | 15.72                      | 16.23            |
> | TIMIT           | 15.83                       | 16.45            |
> | LibriSpeech  | 16.62                      | 17.08            |
> | WSJ             | 15.34                      | 15.47            |
>
> As can be seen, the multi-band architecture consistently outperforms the single-band model, even with a slightly smaller parameter count. Specifically, our multi-band architecture achieves improvements across all datasets, demonstrating that the performance gains are not solely due to model size, but are indeed a result of the multi-band approach itself.
>
>
> \
> **Mamba blocks:**
> \
> Thank you for your suggestion to conduct a comparative ablation study to evaluate the effectiveness of the Mamba (SSM) layers against other architectures, such as transformers, LSTMs, and CNNs. The requested experiments have been conducted, and the results are summarized in the table below. This table reports the average NMSE values obtained, with all models incorporating a nonlinearity term of η = 0.5. Numbers in parentheses represent the parameter count for each model. Each model utilizes three bands: two small sub-bands and one medium-sized band, where the medium band is approximately twice the size of the small bands.
> |             | Transformer (34M) | LSTM (37.5M) | Convolution (41.9M) | Ours (31.9M) |
> |-------------|-------------------|--------------|---------------------|--------------|
> | Factory     | 12.60             | 12.17        | 4.62                | 15.94        |
> | Timit       | 12.90             | 11.83        | 6.57                | 16.36        |
> | WSJ         | 12.04             | 11.88        | 6.43                | 15.32        |
> | Librispeech | 13.86             | 12.99        | 6.80                | 16.95        |
>
> The results demonstrate the superior performance of our Mamba-based multi-band architecture across all datasets compared to the Transformer, LSTM, and convolution-based alternatives. While the Transformer and LSTM approaches provide competitive results, the convolution-based architecture performs less effectively in its current implementation.
>
> \
> **NOAS loss:**
> \
> You are correct that our explanation regarding the NOAS loss formulation (Eq. 12) could have been clearer. We will elaborate on the motivation behind the NOAS optimization in the revised version of the paper which we will upload soon.
> The secondary path S attenuates certain frequencies that the primary path P does not, which presents a challenge during training. Specifically, under the traditional loss function (Eq. 11), the model is penalized for high error signals at these frequencies, even when it has produced an optimal anti-signal. This is because the secondary path S inherently attenuates those frequencies, which results in an unfair penalization of the model.
> Ideally, we would like to avoid punishing the model in these situations.
> The NOAS loss (Eq. 12) mitigates this issue by incorporating S symmetrically on both sides of the NMSE calculation. In this case, if S  nullifies certain frequencies, the error contribution from those frequencies is also nullified in the target. This ensures that the model is not unjustly penalized for frequency bands where the secondary path diminishes the energy.

---

> > ### Author Response · Authors · 2024-11-22
> > **Addressing Review Comments #2**
> >
> > **“The other weakness”:**
> > \
> > We recognize the importance of demonstrating the capabilities of our method in more realistic conditions. To address this concern, we have conducted additional experiments using the dataset you suggested. Specifically, we employed the "Acoustic Path Database for ANC in-ear Headphone Development" (Liebich et al., 2019), which includes acoustic paths from 23 individuals (both primary and secondary paths). We trained our DeepAAC and the baseline methods on the new simulation conditions and evaluated their performance on factory and babble noise from the NoiseX-92 dataset, alongside speech samples from the WSJ dataset.
> > The results, summarized below, present the average NMSE across these categories.
> > |         | ARN   | DeepANC | Ours  |
> > |---------|-------|---------|-------|
> > | Factory | 8.97  | 9.29    | 12.09 |
> > | Babble  | 11.17 | 10.94   | 13.87 |
> > | WSJ     | 10.70 | 8.26    | 12.23 |
> >
> > The results presented in the table demonstrate that our method outperforms the baseline methods across all tested categories. Specifically, our approach achieves an average improvement of 2.8 dB on Factory noise, 2.7 dB on Babble noise, and 1.53 dB on WSJ speech samples. These results underscore the robustness and effectiveness of our method in realistic conditions, where it consistently delivers superior performance in both noise and speech cancellation tasks.
> > We hope this addresses your concern and highlights the generalizability of our proposed method. Thank you for the constructive feedback, which has allowed us to strengthen the evaluation of our work.
> >
> > \
> > **Questions:**
> > \
> > **Q: ”In Section 3.4, could you elaborate…”**
> > \
> > **A:** Thank you for your thoughtful question. Regarding the gradient descent-based algorithm employed for Eq. (11) during the pre-processing stage, we used the Adam optimizer to minimize the NOAS loss specified in Eq. (11). The optimization process started from a randomly initialized signal y′, and we optimized the NOAS loss for approximately 2000 steps. This number of steps was determined empirically, as it consistently led to convergence of the NOAS loss across different scenarios.
> > The learning rate was initially set to 0.01 and was reduced twice by a factor of 0.2 during the optimization process to ensure stable convergence.
> > We appreciate you pointing out the lack of explicit mention of nonlinearity in Eq. (11). We inadvertently omitted the nonlinear term in Eq. (11), and we sincerely thank you for highlighting this oversight. We will correct this in the revised version of the manuscript.
> > As you point out, in the purely linear case, there is no reason why the problem cannot be solved using a direct linear least squares solver.
> > However, the primary focus of our work is on addressing scenarios where nonlinearities are inherent, as they are more representative of real-world ANC tasks. Linear approaches like least squares solvers are not suitable for these cases.
> >
> > \
> > **Q: ”Should the s in Figure 3b…”**
> > \
> > **A:** Thank you for pointing this out. You are correct that the s in Figure 3 should be upper-case S. We will fix this in the revised version of the manuscript. Thank you for bringing this to our attention.
> >
> > \
> > **Q: ”The font size and resolution…”**
> > \
> > **A:** Thank you for your feedback. We agree that the font size and resolution of the plots in Figures 4 and 5 can be improved for better clarity. We will address this and update the figures in the revised version of the manuscript.
> >
> > \
> > **Q: ”Section 4.1, line 347, should…”**
> > \
> > **A:** Thank you for pointing this out. We will fix this in the revised version of the manuscript.
> >
> > \
> > **Q: “For ablation studies…”**
> > \
> > **A:** Yes, "without NOAS optimization" indeed means using only the L_{ANC} loss as defined in Eq. (10). We agree that this was not clearly stated in the current version of the manuscript. We will explicitly clarify the loss function used in the ablation study when NOAS optimization is not applied in the revised version.
> >
> > \
> > **Q: ”In Section 4.3, lines 427-428…”**
> > \
> > **A:** The observation about the "+S-Multiband-NOAS" configuration having lower performance than "+M-Multiband-NOAS" is indeed expected due to the smaller model size (8M vs. 15.8M, as shown in Table 6). However, the reason we compare both is that we utilize both sizes—medium and small—in our multi-band architecture. A description of this setup is provided in Section 4.3 of the paper, where we describe the experiment architecture as employing one medium band and two small subbands.  Additionally, this comparison serves as a sanity check to verify the consistency of the architecture’s behavior with respect to the model’s size and its complexity. Thank you for highlighting this point!

---

> ### Comment · Reviewer_5EHs · 2024-11-24
> **Response to author's rebuttal -- concerns on Mamba (SSM) blocks**
>
> I thank the authors for their responses, which have partially addressed my questions. However, my main concerns for this work still remain, i.e., the motivation and justification for utilizing Mamba (or more generally, state-space model (SSM)) architectures for this specific application in audio processing, namely active audio cancellation (AAC), are still lacking. Even though the authors have provided additional results on comparing Mamba blocks with other architectures (Transformer, LSTM, Convolutional layers) which are appreciated by the reviewer, such information is not sufficient nor thorough enough to justify the advantages of using SSMs for AAC. In the following I will list a few more major concerns and questions after reading the authors' responses.
>
> ### **Main concerns:**
>
> - **Missing details regarding the comparison with other architectures.** In the Mamba vs. {transformer, LSTM, convolution} experiment, only the NMSE values and model sizes are provided, and the detailed setting of each model is missing. For example, number of layers, kernel sizes, uni- or bi-directionality of the compared modules. It is rather difficult to judge if Mamba is actually more effective than other architectures in your application without knowing the details and how you fairly set up the experiment.
>
> - **No complexity comparison.** It is also important to include complexity analysis when *comparing Mamba with transformers*. One of the advantages of SSMs over transformers is that their complexity scales linearly rather than quadratically, thus are much more efficient than transformers, especially in modeling long-dependent sequences/inputs. Therefore, computational cost analysis is equally important as estimation quality comparison, and measurements such as FLOPs (e.g., see [1]) and Real-Time Factor (RTF) (e.g., see [2]) should be reported in addition to NMSE. (Note: processing latency and memory requirements are very important factors to be considered when developing models for audio applications such as ANC (and of course AAC). Surprising, there is no this kind of analysis throughout the paper.)
>
> > [1] Chao, Rong, et al. "An Investigation of Incorporating Mamba for Speech Enhancement." arXiv preprint arXiv:2405.06573 (2024).
>
> > [2] Zhang, Xiangyu, et al. "Mamba in Speech: Towards an Alternative to Self-Attention." arXiv preprint arXiv:2405.12609 (2024).
>
> - **Only short audio samples were considered.** In your studies, only short audios (3 second long) were tested. My concern is in this scenario, the benefits of using Mamba in terms of efficiency might not be obvious over transformers as the inputs are short. To further demonstrate Mamba's advantages for justifying its usage for AAC, the authors should also conduct experiments on much longer audio inputs.
>
> ### **Questions:**
>
> - It is a bit surprising to see that using convolutional layers performs significantly worse than using other modules given the larger model size (41.9M). It is known that Conv layers are good at extracting local information. Given that your inputs are 3-second long only, Conv layers should still be able to work reasonably well. Do the authors have any explanation on the poor performance of Convolution in your experiment?
>
> - The NMSE numbers you show in the new tables seem to be missing a "-" sign?
>
> - No revisions to the manuscript?

---

> ### Author Response · Authors · 2024-11-25
> **Addressing Review Comments #3**
>
> We thank the reviewer for the additional feedback.
>
> >Missing details regarding the comparison with other architectures. In the Mamba vs. {transformer, LSTM, convolution} experiment, only the NMSE values and model sizes are provided, and the detailed setting of each model is missing. For example, number of layers, kernel sizes, uni- or bi-directionality of the compared modules. It is rather difficult to judge if Mamba is actually more effective than other architectures in your application without knowing the details and how you fairly set up the experiment.
>
> We appreciate the reviewer’s comment regarding the experimental setup for comparing Mamba with other architectures and agree that providing these details is important for a fair evaluation. Below, we clarify the configurations used for each model in the comparison:
>
>
>
> **Transformer:** We employed an ARN-based transformer. For the signal sub-bands, a single layer was used, while for the signal full-band, two layers were used. The transformer modules were configured to be bidirectional with d_model of 512.
>
> \
> **LSTM:** For the LSTM architecture, we utilized the torch.LSTM module with two layers for the signal sub-bands and four layers for the signal full-band. All with hidden_size of 256.
>
> \
> **Convolutional Network:** The convolutional architecture comprised 8 layers for all model bands. For the signal sub-bands, we used a kernel size of 1×2×2, while for the signal full-band, the kernel size was 1×2×4, with the last dimension indicating the kernel depth.
>
>
> In all models, we tried to keep a relation of 1:2 between a sub-band module and the full-band module.
> We adopted the same learning rate (1.5e-4) and batch size (2) used in the original Multi-Band Mamba architecture. The learning rate was decayed by a factor of 0.5 every 2 steps following a warm-up period of 30 epochs. Additionally, the encoder modules E0,…, EQ​ , and the decoder D from DeepAAC retained their original dimensions as described in the original architecture.
> All models were trained and evaluated under identical conditions to ensure a fair comparison, including consistent data preprocessing, training hyperparameters, and evaluation metrics. We will include these details in the revised manuscript for better clarity.
> Thank you for pointing this out, and we hope this addresses your concern.
>
>
> >No complexity comparison. It is also important to include complexity analysis when comparing Mamba with transformers. One of the advantages of SSMs over transformers is that their complexity scales linearly rather than quadratically, thus are much more efficient than transformers, especially in modeling long-dependent sequences/inputs. Therefore, computational cost analysis is equally important as estimation quality comparison, and measurements such as FLOPs (e.g., see [1]) and Real-Time Factor (RTF) (e.g., see [2]) should be reported in addition to NMSE. (Note: processing latency and memory requirements are very important factors to be considered when developing models for audio applications such as ANC (and of course AAC). Surprising, there is no this kind of analysis throughout the paper.)
> [1] Chao, Rong, et al. "An Investigation of Incorporating Mamba for Speech Enhancement." arXiv preprint arXiv:2405.06573 (2024).
> [2] Zhang, Xiangyu, et al. "Mamba in Speech: Towards an Alternative to Self-Attention." arXiv preprint arXiv:2405.12609 (2024).
>
>
> We fully agree on the importance of complexity and computational efficiency in evaluating models for audio applications such as ANC and AAC. In response, we comprehensively analyzed our proposed method compared to DeepANC and ARN, focusing on FLOPs and NMSE.
>
> The results are summarized in the table below:
>
> | Method  | FLOPs (G) | NMSE (dB) |
> |---------|-------|--------|
> | Ours    | 2.419 | -13.46 |
> | ARN     | 5.281 | -11.61 |
> | DeepANC | 7.199 | -10.69  |
>
> Our model demonstrates significantly lower FLOPs than DeepANC and ARN, highlighting its computational efficiency. The NMSE is of babble noise with eta=0.5.
> We will include this discussion and table in the revised manuscript to address this important aspect.
> Thank you for highlighting this key point, and we hope the additional data satisfies your concerns.

---

> > ### Author Response · Authors · 2024-11-25
> > **Addressing Review Comments #4**
> >
> > > Only short audio samples were considered. In your studies, only short audios (3 second long) were tested. My concern is in this scenario, the benefits of using Mamba in terms of efficiency might not be obvious over transformers as the inputs are short. To further demonstrate Mamba's advantages for justifying its usage for AAC, the authors should also conduct experiments on much longer audio inputs.
> >
> > Thank you for your insightful comment regarding the testing of longer audio samples. To address your concern, we conducted additional experiments using 12-second audio signals from the NoiseX-92 dataset. These experiments demonstrate that our proposed method, referred to as "FULL METHOD" in the paper (1M full-band + 2S sub-bands + NOAS), maintains its performance over longer audio durations. The results, shown below, indicate minimal variation in the performance compared to the previously reported results on 3-second samples (eta=0.5).
> >
> > |Noise Type | Full Method (12s) | Full Method (3s) |
> > |---------|-------|--------|
> > |Factory | 	-16.11 | 	-16.25 |
> > |Babble | 	-20.42 |	-20.17 |
> >
> >
> > This consistency underscores Mamba's robustness and efficiency, even for longer input durations, further justifying its usage for AAC. We appreciate your suggestion and hope this additional evidence strengthens the argument for our approach.
> >
> >
> > **Questions:**
> > > It is a bit surprising to see that using convolutional layers performs significantly worse than using other modules given the larger model size (41.9M). It is known that Conv layers are good at extracting local information. Given that your inputs are 3-second long only, Conv layers should still be able to work reasonably well. Do the authors have any explanation on the poor performance of Convolution in your experiment?
> >
> > We appreciate the reviewer’s observation regarding the performance of the convolutional model despite its larger size. For this experiment, we employed a convolutional autoencoder architecture based on the skeleton of DeepANC. Specifically, it consisted of 4 encoder layers and 4 decoder layers, with batch normalization applied after each layer to stabilize training and improve convergence. While this setup is effective at extracting local information, it represents a relatively "vanilla" CNN architecture akin to those popular around 2015. Consequently, it does not incorporate the significant advancements in CNN design made in recent years. The kernel size for the signal sub-bands was 1×2×2, while for the signal full-band, the kernel size was 1×2×4, with the last dimension indicating the kernel depth.
> >
> > We acknowledge that better hyperparameter tuning and architecture adjustments could potentially improve its performance and make it more competitive. Thank you for bringing this to our attention.
> >
> >
> > > The NMSE numbers you show in the new tables seem to be missing a "-" sign?
> >
> > Thank you for pointing this out. Yes, the NMSE values in the new tables are indeed missing the "-" sign. We apologize for the oversight and will correct it in the revised version of the manuscript.
> >
> >
> > > No revisions to the manuscript?
> >
> > Thank you for your observation. We prioritized conducting additional experiments and deferred the textual revisions to a later stage. Currently, we are in the final stages of revising the manuscript and will address all reviewer comments comprehensively before submission.

---

> ### Comment · Reviewer_5EHs · 2024-11-27
> **Official comment by Reviewer 5EHs**
>
> I sincerely thank the authors for the additional experiments on complexity analysis. However, I still remain unconvinced about the motivation and advantages of using Mamba in this specific application. It is appreciated that the proposed DeepAAC architecture seems to achieve the best efficiency compared to existing ARN and DeepANC methods in your experiment (with FLOPs results), which is definitely great, but I found it insufficient to infer from the results that Mamba is the better choice than other architectures as the building blocks for AAC/ANC networks. That being said, an ablation study that compares among using different building blocks for the $B_0,\dots,B_Q$ in Figure 1 or something similar would have been more helpful. Given that AAC is a very specific problem in audio applications, I feel that a much stronger motivation and comprehensive evidence for why Mamba is the go-to choice to the problem are very necessary. Otherwise, it appears to be a straightforward replacement of existing architectures by the Mamba blocks, and thus the machine learning contributions are somewhat limited given the specificity of the problem. Based on that, I've decided to keep my initial score unchanged.

---

> > ### Author Response · Authors · 2024-11-27
> > **Addressing Review Comments #5**
> >
> > We sincerely appreciate your detailed feedback and thoughtful critique regarding our work. In response to your suggestion about comparing different building blocks for $B0,...,BQ$ in Figure 1, we would like to clarify that we have already conducted such an ablation study (see the 2nd table under "Addressing Review Comments #1").
> > Our experiments explore the use of alternative building blocks and demonstrate that Mamba consistently achieves better results in terms of FLOPs and cancellation performance for the AAC/ANC problem.
> > We would like to ask for additional clarity on what specific evidence or analysis would help solidify the motivation and advantages of Mamba in your view. Are there specific scenarios, metrics, or comparisons you'd like to see?

---

### Official Review · Reviewer_Lmew · 2024-11-04

**Soundness:** 3
**Presentation:** 3
**Contribution:** 3
**Rating:** 6
**Confidence:** 4

**Summary:**

A novel deep learning approach to Active Audio Cancellation (AAC) is proposed based on the multi-band Mamba architecture. The paper claims that it is a more broadly defined problem than the traditional ANC problem as it can handle various other sounds. The proposed method also employs a two-stage training paradigm to overcome the nonlinearity of the ANC path. The performance appears to be good based on the simulated experiments.

**Strengths:**

- The multiband Mamba architecture appears to be a reasonable choice for the task.
- Ablation tests and other experimental designs (including the choice of datasets) are well thought-out.
- The proposed method improves the other baseline methods.

**Weaknesses:**

- Although it is a reasonable choice, the proposed multiband Mamba architecture appears to be a straightforward extension of existing architecture into a multiband version. Algorithmic contributions are thus limited.
- It is not clear why the second stage loss function is needed, if the secondary path filter and loudspeaker processing part is differentiable.
- All experiments are based on image method-based room simulation. Its effectiveness on real-world test signals has not been reported.
- The system takes in 3 sec of audio during training. It is not reported what's the delay of the system during the inference. Since most of the ANC applications require real-time processing, this delay might be critical for the proposed method to be useful.
- Some sections are written without any line breaks, seriously harming readability.

**Questions:**

- It seems that the second loss function is to improve the model's performance after doing the first round, where the error is supposed to be reduced. The authors said that the secondary path S and the loudspeakr function f_LS is nonlinear, making the algorithm effectively update the Deep AAC model. However, these two functions must be differentiable anyway, so they were part of the inference in the first round. Then, the first round must be able to go through them and update the model anyway? Specifically, S is just a convolution filter acquired from the room simulation? So it must be differentiable? If so, isn't eq 12 solely about f_LS rather than f_LS AND S?

- I think it's an overclaim that what this paper aims to do is a totally different task than ANC. Although ANC, as its name suggests, might be about reducing noise, many systems are trying to cancel "any" sound that's coming into the system, as far as I know.

---

> ### Author Response · Authors · 2024-11-22
> **Addressing Review Comments #1**
>
> We appreciate the reviewer’s feedback and we would like to clarify.
>
> \
> **Weaknesses:**
> \
> **W1: Although it is a reasonable choice, the proposed multiband Mamba architecture appears to be a straightforward extension of existing architecture into a multiband version. Algorithmic contributions are thus limited.**
> \
> **A1:** The multi-band extension of the Mamba architecture was a deliberate choice to address the challenges posed by higher-frequency components, which are common in speech signals. Speech cancellation was one of the main focuses of our work.
> The effectiveness of the multi-band architecture is further underscored through an additional experiment we conducted where we compared its performance against a single-band approach of comparable model size. Specifically, we evaluated a large single-band model with 34M parameters and a multi-band model with 31.9M parameters (using 3 bands). Both models were tested on Factory noise from NoiseX-92 as well as speech datasets from TIMIT, LibriSpeech, and WSJ.
> The results, summarized in the table below, report the average NMSE values for each dataset with a nonlinearity term of η = 0.5.
> | Dataset        | Single-Band (34M) | Multi-Band (31.9M) |
> |-----------------|----------------------------|------------------|
> | Factory         | 15.72                      | 16.23            |
> | TIMIT           | 15.83                       | 16.45            |
> | LibriSpeech  | 16.62                      | 17.08            |
> | WSJ             | 15.34                      | 15.47             |
>
> These results demonstrate that our multi-band approach consistently outperforms the single-band model, even when the single-band model has slightly more parameters. Importantly, this improvement is observed across all datasets, whether they involve noise or speech.
> Additionally, we believe an important algorithmic contribution of our work is the introduction of the NOAS optimization, which enhances performance across different variations of our method, as shown in Table 3. We believe that the benefit of this optimization lies not only in its application to our method but also in its broader potential to improve performance in any supervised learning-based ANC approach.
>
> \
> **W2: It is not clear why the second stage loss function is needed if the secondary path filter and loudspeaker processing part are differentiable.**
> \
> **A2:** The key issue arises from the behavior of the secondary path filter S, which attenuates certain frequencies. In contrast, the primary signal P may contain energy in these frequencies. Under the traditional training loss function (Eq. 11), this results in the model being penalized for high error signals at frequencies where the secondary path filter S heavily attenuates the signal, even though the model may have generated an optimal anti-signal. This mismatch occurs because the error calculation is influenced by the primary signal's energy at those frequencies, despite S effectively nullifying them.
> To address this issue, we introduce the NOAS loss (Eq. 12), which incorporates the secondary path filter S symmetrically on both sides of the NMSE calculation. This design ensures that if S attenuates certain frequencies, the corresponding error contribution is nullified in the target signal. In other words, the model is not unjustly penalized for errors at frequencies where the secondary path diminishes the energy. This modification is critical to accurately reflecting the performance of the model in the presence of non-ideal acoustic conditions.
>
> \
> **W3: All experiments are based on image method-based room simulation. Its effectiveness on real-world test signals has not been reported.**
> \
> **A3:** As the reviewer suggested, we have extended our evaluation to real-world test signals. Specifically, we utilized the "Acoustic Path Database for ANC in-ear Headphone Development" (Liebich et al., 2019), which includes acoustic paths from 23 individuals, encompassing both primary and secondary paths.
> We applied DeepAAC, along with baseline approaches, to the updated simulation conditions and assessed their performance using factory and babble noise from the NoiseX-92 dataset, in addition to speech samples from the WSJ dataset.
> The following results present the average NMSE across these categories.
> |         | ARN   | DeepANC | Ours  |
> |---------|-------|---------|-------|
> | Factory | 8.97  | 9.29    | 12.09 |
> | Babble  | 11.17 | 10.94   | 13.87 |
> | WSJ     | 10.70 | 8.26    | 12.23 |
>
> Evidently, our method outperforms the baseline approaches across all evaluated categories, even when applied to real-time measured paths. In particular, our method demonstrates an average improvement of 2.8 dB for Factory noise, 2.7 dB for Babble noise, and 1.53 dB for WSJ speech samples. These new results demonstrate the robustness and efficacy of our approach in real-world scenarios in addition to image-method based simulation methods.

---

> ### Author Response · Authors · 2024-11-22
> **Addressing Review Comments #2**
>
> **W4: The system takes in 3 sec of audio during training. It is not reported what's the delay of the system during the inference. Since most of the ANC applications require real-time processing, this delay might be critical for the proposed method to be useful.**
> \
> **A4:** We appreciate the reviewer for raising this important point regarding the system's delay during inference. As of now, we have not specifically evaluated the inference delay of our method. However, we acknowledge that real-time processing is a critical requirement for ANC applications. We plan to explore optimization techniques such as pruning, quantization, and other runtime optimizations to reduce the system's inference delay.
>
> \
> **W5: Some sections are written without any line breaks, seriously harming readability.**
> \
> **A5:** We thank the reviewer for pointing out the readability issue. We will ensure that this is addressed and fix the formatting in the revised version to improve readability.
>
>
> \
> **Questions:**
> \
> **Q1: It seems that the second loss function is to improve the model's performance after doing the first round, where the error is supposed to be reduced. The authors said that the secondary path S and the loudspeaker function f_LS is nonlinear, making the algorithm effectively update the Deep AAC model. However, these two functions must be differentiable anyway, so they were part of the inference in the first round. Then, the first round must be able to go through them and update the model anyway? Specifically, S is just a convolution filter acquired from the room simulation? So it must be differentiable? If so, isn't eq 12 solely about f_LS rather than f_LS AND S?**
> \
> **A1:** The secondary path S attenuates certain frequencies that the primary path P does not address, which introduces a challenge in the training process. Specifically, when using the traditional loss function (Eq. 11), the model can be penalized for high error signals in these frequencies, even if it has effectively generated an optimal anti-signal. This occurs because S inherently suppresses those frequencies, leading to energy in the error signal. Thus, the NOAS optimization loss function is designed to account for this discrepancy by applying S on both sides of the NMSE, so whenever \( S \) nullifies certain frequencies, the error contribution from those frequencies is also nullified in the target. We will clarify it in the revised version of the paper.
>
> Regarding the comment, "Eq. 12 solely about f_LS​ rather than f_LS AND S": we chose to include the S projection in Eq. 12 to leverage the prior knowledge embedded in the model, which was already trained to optimize the S-projected component of its output. Empirically, we observed that this approach yields better results compared to directly optimizing over y as it demonstrated in Table 7 in the paper. As shown, the NMSE distance between y and y∗ is significantly greater than the distance between P∗x and S∗y, emphasizing the effectiveness of optimization over S*y. To provide visual intuition for this phenomenon, we have included Figure 3 in the paper.
>
> \
> **Q2: I think it's an overclaim that what this paper aims to do is a totally different task than ANC. Although ANC, as its name suggests, might be about reducing noise, many systems are trying to cancel "any" sound that's coming into the system, as far as I know.**
> \
> **A2:** We apologize for the overstatement in our terminology and appreciate the reviewer’s feedback. After reviewing the existing literature, we found that most ANC systems focus primarily on canceling noise, such as factory engine sounds, and not speech. That is why we initially referred to our method as "Active Audio Cancellation." However, we understand the need for more precise terminology. In the revised version, we will adjust this term and references to use a more modest term that better reflects the scope of our work.

---

> > ### Comment · Reviewer_Lmew · 2024-11-26
> > **Thanks for the feedback**
> >
> > I appreciate the authors' additional efforts in testing the model on another dataset. While I understand the difference between primary and secondary paths, it's still unclear to me why the optimization has to be staged in this way. That being said, the algorithm is still far from real-time processing, which is a deal breaker for the ANC applications. So, I would like to stay with my original score.

---

### Author Response · Authors · 2024-11-28
**Overall Author Response**

We thank all the reviewers for their valuable time and constructive feedback on our submission. We have carefully considered all comments and suggestions, addressing them in detail and reflecting them in a revised version of our paper:


1. To reduce potential overclaims about Active Audio Cancellation, we refined the phrasing in the abstract and introduction sections.

2. We have also clarified the motivation for the NOAS optimization by adding a detailed explanation, aiming to make it more comprehensible.

3. Regarding Reviewer 5EH's concerns about the architecture choices, we included the performance of a single large band in Table 5 to highlight the advantages of the multi-band approach. Additionally, we added Table 10 (presented in Appendix A.2) to demonstrate the impact of the Mamba block through an ablation study.

4. To address concerns about real-world performance and generalizability, we conducted experiments using real-world measured paths, which are now detailed in the newly added "Real-Wold Simulation" section (4.3).

5. In response to comments on model complexity, we added a FLOPs comparison in Table 7 to better illustrate the efficiency of our approach.


We appreciate the opportunity to improve our submission and are happy to address any further questions or concerns during this discussion period.

---

### Note · Authors · 2025-05-25

I have read and agree with the venue's withdrawal policy on behalf of myself and my co-authors.

---

### Meta-Review · Area_Chair_riFw · 2024-12-12

**Metareview:**

While appreciating the technical novelty of the proposed method, the reviewers have spotted many weaknesses of the paper. In particular, Reviewer 5EHs, who strongly suggests rejection even after extensive author-reviewer discussions, point out the motivations and justifications need more scientific rigor. Considering the standard of this conference and the limited budget of my batch, I would recommend rejecting the paper. Saying that, the authors are encouraged to continue the work.

**Additional Comments On Reviewer Discussion:**

The authors’ rebuttal has partially addressed the reviewers’ comments. However, the following concerns remain unsolved:

1. Reviewer 5EHs, who strongly suggests rejection even after extensive author-reviewer discussions, point out the motivations and justifications need more scientific rigor. Considering the standard of this conference and the limited budget of my batch, this weakness is fatal.

2. Reviewer Lmew, who weakly suggests acceptance, still complains about the computational efficiency of the proposed method.

3. Reviewer m3g7, who weakly suggests acceptance, still complains about reproducibility of the proposed technique.

---

### Decision · Program_Chairs · 2025-01-22

Reject